# Preparation, Mechanical Properties, and High-Temperature Wear Resistance of Ti–Al–B alloy

**DOI:** 10.3390/ma12223751

**Published:** 2019-11-14

**Authors:** Gongjun Cui, Yanping Liu, Guijun Gao, Huiqiang Liu, Sai Li, Ziming Kou

**Affiliations:** 1College of Mechanical and Vehicle Engineering, Taiyuan University of Technology, Taiyuan 030024, China; yanpingliutyut@sina.com (Y.L.); guijungao333@163.com (G.G.); huiqiang1990@163.com (H.L.); lisaityut@163.com (S.L.); kzmingtyut@163.com (Z.K.); 2National-local Joint Engineering Laboratory of Mining Fluid Control, Taiyuan 030024, China

**Keywords:** titanium alloys, Boron, high-temperature alloying, wear resistance, friction

## Abstract

Ti–Al matrix alloy reinforced with a high content of boron was fabricated by using a high-temperature alloying method and powder metallurgy technique (P/M). The preparation method of Ti–Al–B alloying powder was put forward. Phases, microstructure, and mechanical properties of the alloys were investigated. Wear and friction performance were studied by using a ball-on-disc tribotester sliding against a Si_3_N_4_ ceramic ball from 23 °C (room temperature) to 900 °C. The Ti–Al–B alloy had a higher specific strength than that of the Ti–Al alloy. The boron element obviously enhanced the wear resistance and mechanical properties of the alloys because of the formation of borides (TiB_2_ and AlB_2_) in matrices and the stable oxide film on the wear tracks. Friction coefficients of alloys were independent of the boron element. The wear mechanisms of the alloys transferred from fatigue wear to oxidative wear with the increase in temperature.

## 1. Introduction

Titanium alloys are characterized by low density, high strength, and oxidation resistance; they have been widely used as high-temperature mechanical components such as nozzles, divergent flaps and blades in aviation industries, and braking systems [1,2,3,4,5,6]. The mechanical parts could suffer from severe wear at high temperatures. Therefore, it is necessary to study the wear and friction properties of titanium matrix alloys at high temperatures. Although investigations of tribological properties of Ti matrix materials have been done, alloys showed low wear resistance [7,8]. This will limit the application of Ti alloys. When the power of engines further increases, titanium matrix alloys must possess more excellent high-temperature wear resistance.

Ceramic particles were used to strengthen the wear resistance of titanium alloys such as TiC, TiB_2_, TiN, and SiC [9,10,11,12]. The prepared titanium matrix composites containing ceramic particles showed an obvious wear resistance because the ceramic particles in the matrix could sustain the applied load and restrain dislocation during the friction process. Additionally, the ceramic particles enhanced mechanical properties of materials due to the dispersion strengthening effect. However, the ceramic particles scraped the contact surfaces of tribo-couples, resulting in the increase of friction coefficients of composites with increasing ceramic particle content [12,13]. Boron-reinforced alloys were reported by many researchers. Fu et al. [14] studied the wear resistance of Fe_83_B_17_ alloys at room temperature (RT). The nanostructure obviously strengthened the wear resistance of the Fe_83_B_17_ due to the greater Fe_2_SiO_4_ layer on the contact surfaces. Wu et al. [15] prepared a boronized layer on Ti6Al4V by using a laser alloying technique. The hardness of the coating greatly increased because of the formation of TiB and TiAl compounds, and the boronized coating had better wear resistance than that of Ti6Al4V. Dixit et al. [16] modified the wear resistance of Ti6Al4V with boron at 200 °C. They found that boron had a refining effect on grain size. However, TiB showed a malign effect on the wear performance of Ti6Al4V because of the delamination of TiB particles. The addition of solid lubricants was another method to reinforce the wear resistance of alloys at high temperatures, including the use of low-temperature and high-temperature solid lubricants [12,17,18,19,20]. Different types of Ti matrix materials containing two or three solid lubricants were reported. A lubricating film was formed on the wear tracks under some special load and temperature conditions; this could provide lubricating effects for composites at high temperatures. Shi et al. [21] prepared self-lubricating TiAl matrix composites with Ag+Ti_3_SiC_2_+BaF_2_/CaF_2_. The silver, fluorides, and oxides formed a lubricating film on the wear tracks in order to decrease the friction coefficients of composites (0.45) below 600 °C. Ti_3_SiC_2_ is getting more and more attention in the tribological fields [17,21,22]. On one hand, Ti_3_SiC_2_ directly forms lubricating film on the worn surfaces of TiAl alloys; on the other hand, Ti_3_SiC_2_ decomposes during the friction process to form oxide film, which consists of SiO_2_ and TiO_2_. Although the solid lubricants can decrease the friction and wear of titanium matrix composites at elevated temperatures, the solid lubricants destroy the mechanical properties of materials [18,19,21,23]. Hence, we make an attempt to balance the mechanical properties and friction coefficient as well as wear of titanium alloys at elevated temperatures.

Titanium alloys were important aero materials because of their high specific strength. In view of this, the boron element was used as the alloying element to enhance the mechanical and tribological properties at elevated temperatures. Phases, microstructure, mechanical, and tribological properties of alloys were investigated. Bulk titanium matrix alloys were fabricated by using a powder metallurgy technology which ensured the complete alloying of Ti, Al, and the boron element during a long sintering process. The friction and wear properties were tested on a ball-on-disc tribotester rubbing against a Si_3_N_4_ ceramic ball from 23 °C (room temperature) to 900 °C. The friction and wear mechanisms were discussed.

## 2. Experimental Process

Ti (aladdin, 99.5% purity, Shanghai, China), Al (aladdin, 99.4% purity), and B (aladdin, 99.99% purity) powders were chosen as the starting materials in this study. The sizes of Ti, Al, and B powders were about 65, 80, and 40 μm, respectively. Ti–Al and Ti–Al–B alloying powders were prepared in our lab according to Table 1. Ti–Al alloy was denoted as T, and Ti–Al–B alloy was denoted as TB. The preparation process was as follows: the corresponding powders were well mixed together by using a planetary ball mill. The mixed powders were heated to 1100 °C for 40 min in a vacuum furnace. The alloying powders were ground in a high-energy mill machine for 12 h. The size of the prepared alloying powder was about 10 μm (see Figure 1). A graphite die (inner diameter: 30 mm) was used to fabricate the bulk alloys. Samples were sintered by using a powder metallurgy technique in a vacuum hot-pressing furnace when the vacuum degree reached up to 10^−2^ Pa. Samples were sintered at 1200 °C under a pressure of 3.5 × 10^−7^ Pa for 45 min. After being sintered, the samples were taken out from the furnace and cut into appropriate sizes.

The wear and friction properties were evaluated by using a ball-on-disc tribotester sliding against a Si_3_N_4_ ball in air from 23 to 900 °C. A ball with a diameter of 6 mm and hardness of 15 GPa was fixed. Discs were made of prepared alloys with a size of Φ 30 mm × 4 mm. The sliding radius was 5 mm, and the other parameters were as follows: sliding speed of 0.20 m/s and normal load of 10 N. Selected temperatures were 23 °C (room temperature), 200 °C, 400 °C, 600 °C, 800 °C, and 900 °C, respectively. Each experimental point was repeated three times in order to ensure the accuracy of the experimental data. Friction coefficients of specimens were recorded by computer software.

Microstructures, element distribution maps, and wear morphologies of prepared alloys were analyzed by using scanning electron microscopy (SEM, IT-300, JEOL, Tokyo, Japan) and energy dispersive spectroscopy (EDS, X-MAX-50, Oxford, UK). The densities of the alloys were tested according to Archimedes’ method. The porosity was calculated by the following formula:(1)$$P=\frac{\rho_{o}-\rho }{\rho_{o}}$$
where ***ρ_o_*** was the theoretical density and ***ρ*** was the actual density of the alloys. The phases of the alloys were detected by XRD (Diffractometer-6000, Shimadzu, Kyoto, Japan). The Vickers hardness was tested by a Vickers hardness tester (South Beam Electronics, Suzhou, China, load: 300 g, time: 10 s). The average value of fifteen tests of hardness was reported. For the three-point bending tests, the speed of the crosshead was 3.4 × 10^−4^ s^−1^, and the dimensions of the specimens were 3 mm × 3 mm × 20 mm. Quasi-static uniaxial compressive tests were conducted with a size of Ø 3 mm × 5 mm. The above tests were repeated three times. The cross-sectional area of the wear tracks was evaluated by using a contact surface profiler. The wear rate of the obtained alloys was the total wear volume (mm^3^) divided by the normal load (N) and sliding distances (m), and the unit was mm^3^/N·m. The formula was:(2)$$w=\frac{V}{F•S}.$$

## 3. Results and Discussion

### 3.1. Microstructure and Mechanical Properties of Alloys

Figure 2 shows the XRD patterns of specimens. The main phases of specimen T are Ti_3_Al and TiAl. This indicates that the Ti element fully reacts with the Al element due to the high-temperature solid solution reaction. At high temperatures, N_2_ reacts with Ti and Al elements, forming Ti_2_AlN as a secondary phase of the alloy. Some corresponding peaks are detected in the figure. The Ti_2_AlN phase belongs to the MAX phase which has high-temperature plasticity, a high elasticity modulus, and layer structure [24,25]. The Ti_2_AlN phase can improve the toughness and tribological properties of titanium alloys to some degree. With the addition of boron, Ti and Al elements react and form Ti_2_B and Al_2_B phases that reinforce the mechanical properties and wear resistance of the alloy. According to the XRD result, it is confirmed that the phases of TB consist of Ti_3_Al, TiAl, Ti_2_AlN, TiB_2_, and AlB_2_. And therefore, the reactions are as follows:
4Ti + 2Al → TiAl + Ti_3_Al(3)
4Ti + 2Al + N_2_ → 2Ti_2_AlN(4)
Ti + Al + 4B → TiB_2_ + AlB_2_.(5)

Microstructure and element distribution maps of specimen T are given in Figure 3. A crack or hole is not found on the surface. The microstructure of T is compacted. The Ti or Al element does not agglomerate, according to the EDS results (see Figure 3b,c). This means that the distribution of TiAl compounds is uniform in the matrix (see Figure 2). Figure 4 shows the microstructure and element distribution maps of TB. The microstructure of TB is different from that of T due to the addition of boron. The light gray area is the Al-rich phase which includes AlB_2_ and Ti(Al) phases. The B-rich area is mainly located in the gray area. According to the results of XRD, the TiB_2_ phase is located in the gray area. The TiB_2_ phase forms a network in the matrix.

Table 2 lists the density, porosity, and mechanical properties of the alloys. The hardness (Hv) of TB is higher than that of T due to the strengthening effect of the B element [10,12]. The compressive strength of TB is about 873 MPa, and that of T is 356 MPa. The compressive strength of TB is enhanced by 1.4 times. Meanwhile, the bending strength of TB is increased by about 0.7 times. This means that the strengthening effect of B is very significant to the mechanical properties of the Ti–Al alloy. The density of TB is lower than that of T. It is easy to explain that the density of the alloy decreases because of the addition of low-density boron. The porosity proves the compactness of the specimens (see Figure 3 and Figure 4).

### 3.2. High-Temperature Tribological Properties

Figure 5 gives the trend of friction coefficients of the alloys with different temperatures at 0.20 m/s and 10 N sliding against the Si_3_N_4_ ball. The friction coefficient of T and TB generally decreases as the temperature reaches up to 900 °C from RT. T and TB show a similar value for the friction coefficient. This means that the friction coefficient is independent of boron. However, the temperature greatly influences the friction coefficients of specimens. Figure 6 shows the vibration of specific wear rate of the alloys with different temperatures sliding against the Si_3_N_4_ ball. The wear rates of alloys gradually decrease with increasing temperature. The wear rates are in the range of 0.8−4.2 × 10^−4^ mm^3^/N.m from 23 to 900 °C. The specimens T and TB show a similar value of wear rate when the temperature is below 400 °C. However, the specimen TB shows smaller wear rates than those of specimen T from 400 to 900 °C. There is no doubt that the boron element plays an important part in reducing the wear rates of the Ti–Al matrix alloy at high temperatures. The friction and wear mechanisms will be discussed below.

The friction coefficients and wear rates of alloys fluctuate with testing temperatures. When the temperature is low, it is difficult for a small amount of oxides to form stable oxide film on the contact surfaces of specimens [26,27]. At elevated temperatures, metallic oxides increase (see Figure 7). The speed of oxidation is greater than that of oxide removal. So, large amounts of oxides are compacted in order to form more compacted and continuous oxide film on the wear tracks as the temperature rises (see Figure 8). The oxide film protects materials from oxidation at high temperatures. Additionally, the oxide film separates the contact surfaces of counterparts during sliding, adding the lubricating effect. The high coverage of oxide film leads to low wear rates and friction coefficients [28,29]. Consequently, the friction coefficient and wear rate of the alloys decreases as testing temperature increases. Hardness is an important parameter that obviously influences wear resistance of materials [11,30,31]. The formation of TiB_2_ and AlB_2_ phases increases the mechanical properties of the alloy. Meanwhile, the network TiB_2_ phase effectively sustains the external load during friction. The relatively soft phases in the matrix avoid severe wear [11]. Secondly, material with high deformation resistance easily forms stable oxide film on the wear tracks at high temperatures due to the high hardness. Thus, TB has better friction and wear performance than those of T at testing temperatures.

### 3.3. Wear Surface Analysis

Figure 9 gives the SEM images of worn surface morphologies of T at different temperatures sliding against the Si_3_N_4_ ball. The wear track of the specimen is relatively smooth at room temperature (see Figure 9a). Much wear debris is noted on the wear track. The Ti–Al alloy has a hard and brittle nature. Due to the effect of alternate stress, the material peels off from the contact surfaces and becomes tiny wear debris as the third body [32]. This indicates that T shows fatigue wear at room temperature. At 600 and 900 °C, oxide film and a slight groove are found on the wear tracks of the alloys (see Figure 9b,c). The oxidation reaction of the elements and wear debris is sped up. These oxides form stable oxide film on the wear tracks. The oxide film becomes more and more compacted and continuous on the surfaces with the increase of temperature, resulting in a lower wear rate and friction coefficient of the alloys at high temperatures. The main wear mechanism of T is oxidative wear at 600 and 900 °C.

SEM images of worn surfaces of TB at RT, 600, and 900 °C are given in Figure 10. The morphology is similar to that of T at room temperature (see Figure 10a). The wear mechanism of TB is fatigue wear. When the testing temperatures are 600 and 900 °C, the coverage of oxide film increases compared with that of T at the same temperature. The applied load can destroy the oxide film and hinder the formation of oxide film. After losing the protective effect of oxide film, the fresh surface is exposed on the wear tracks and suffers from wear. Therefore, T shows higher wear rates at high temperatures. TB has better mechanical properties than those of T, so TB has a high deformation resistance in order to form a stable oxide film on the worn surfaces, resulting in the low friction coefficients and wear rates of TB. The main wear mechanism of TB is oxidative wear at 600 and 900 °C.

Figure 11 presents the wear scars of Si_3_N_4_ ceramic balls sliding against TB at RT, 600, and 900 °C. The transferred layer is noted on the wear scars of ceramic balls. With the increase of temperature, the transferred layer becomes compacted and continuous. The composition of the transferred layer is the same as that of the oxide film on the wear tracks, according to the EDS analysis (see Figure 12). The oxide film and transferred layer change the friction mode of tribo-couples, that is, the friction mode of alloy-to-ball turns into internal friction between oxide films. It suggests that the oxide film plays an important part in reinforcing the tribological properties of alloys at high temperatures.

## 4. Conclusions

(1) Ti–Al matrix alloying powders were fabricated by using a high-temperature alloying method. The bulk Ti–Al matrix alloys with a high content of boron were prepared by using a powder metallurgy technique. The microstructure of the alloys was compacted. The phases of the Ti–Al–B alloy consisted of Ti_3_Al, TiAl, Ti_2_AlN, TiB_2_, and AlB_2_. The specimen with boron had better mechanical properties than those of the specimen without boron.

(2) The friction coefficients of the alloys were insensitive to the boron element. The wear resistance of TB was enhanced due to the formation of TiB_2_ and AlB_2_. The network TiB_2_ could carry the applied load during sliding, resulting in the formation of stable oxide film on the wear tracks at higher temperatures. Additionally, the testing temperature obviously influenced the tribological properties of alloys. Ti–Al–B alloy has better tribological properties than those of the Ti–Al alloy from RT to 900 °C.

(3) The wear mechanism of the alloys was fatigue wear at low temperatures. The specimens showed oxidative wear at elevated temperatures.

## Figures and Tables

**Figure 1 materials-12-03751-f001:**
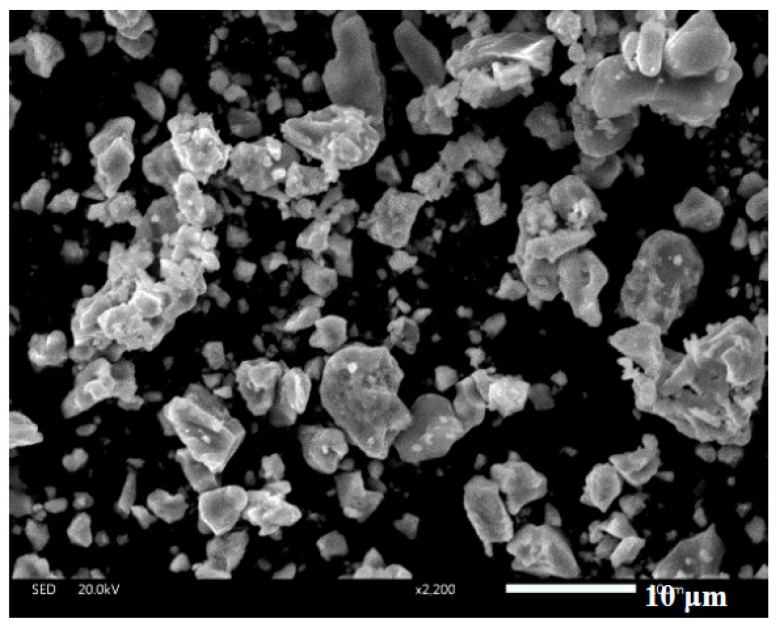
SEM image of alloying powder.

**Figure 2 materials-12-03751-f002:**
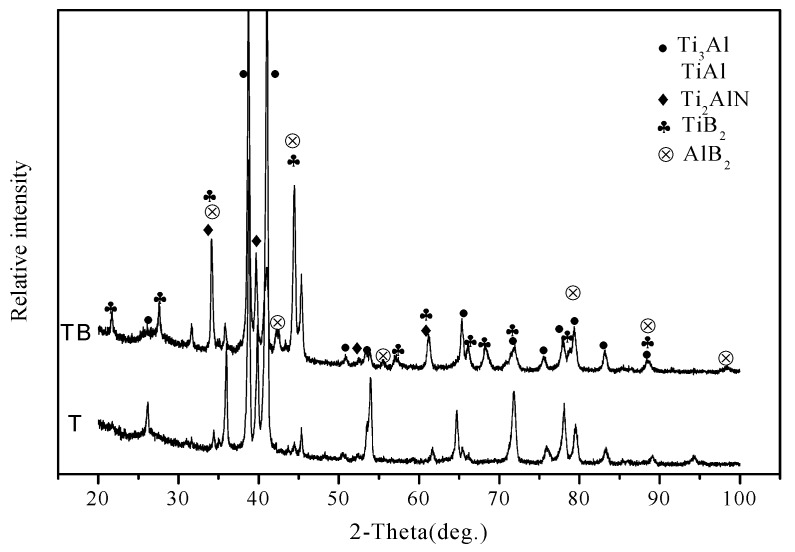
XRD patterns of alloys.

**Figure 3 materials-12-03751-f003:**
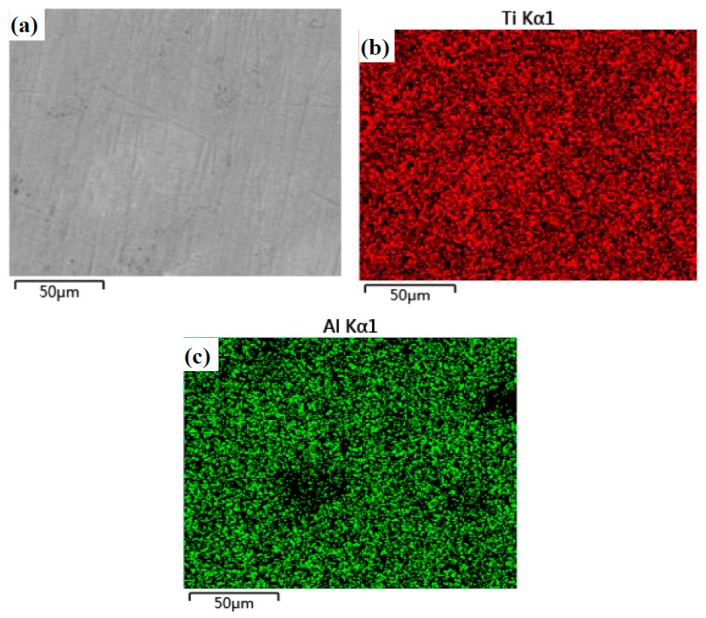
Microstructure and EDS element distribution maps of the Ti–Al alloy (T): (**a**) Microstructure, (**b**) Ti and (**c**) Al.

**Figure 4 materials-12-03751-f004:**
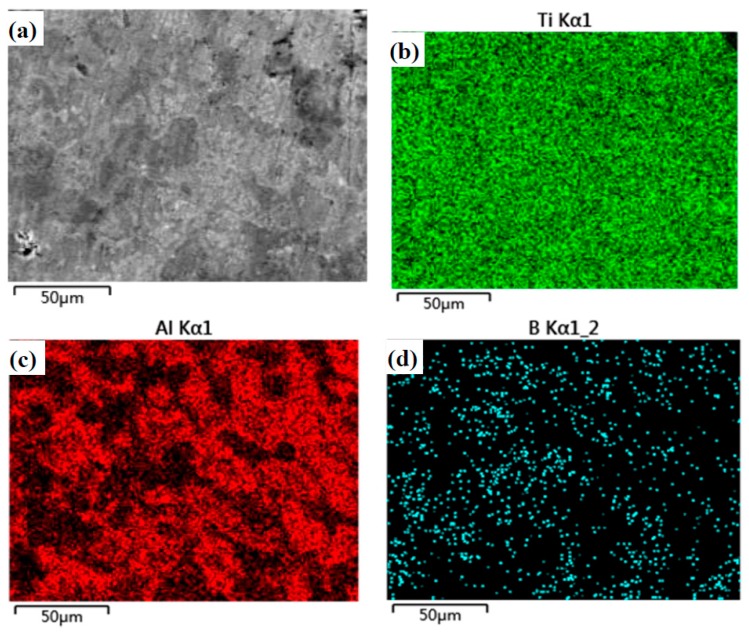
Microstructure and EDS element distribution maps of the Ti–Al–B alloy (TB): (**a**) Microstructure, (**b**) Ti, (**c**) Al and (**d**) B.

**Figure 5 materials-12-03751-f005:**
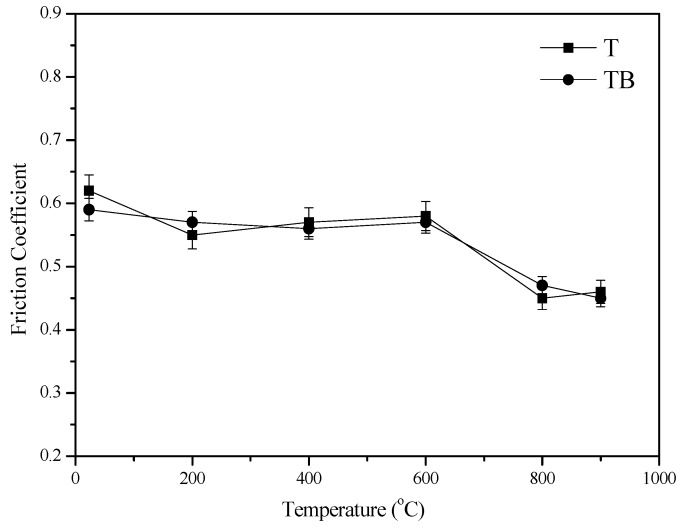
Vibration of friction coefficients of alloys with temperature at 10 N and 0.20 m/s.

**Figure 6 materials-12-03751-f006:**
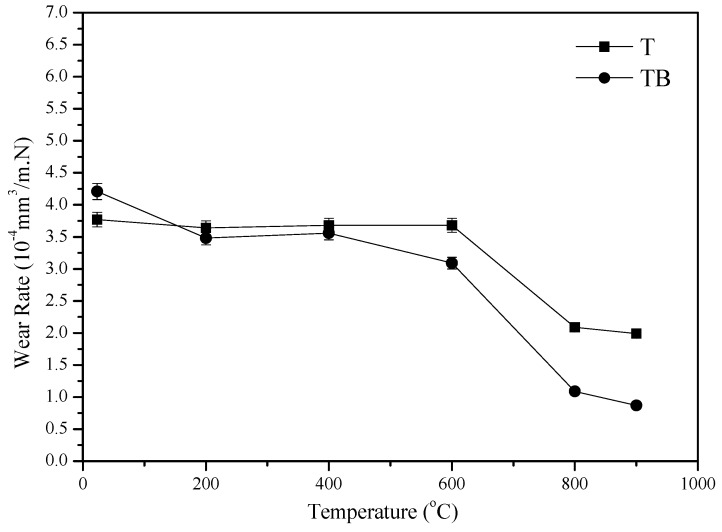
Specific wear rates of alloys with temperature at 10 N and 0.20 m/s.

**Figure 7 materials-12-03751-f007:**
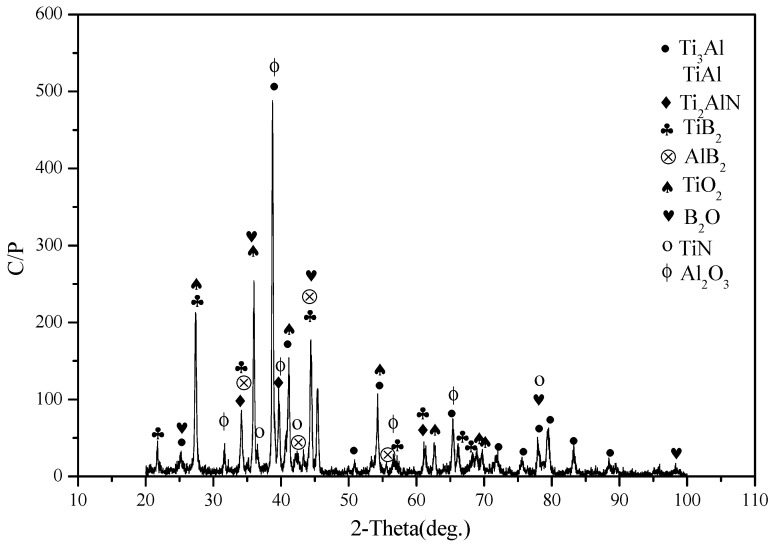
XRD pattern of the wear track of specimen TB at 900 °C.

**Figure 8 materials-12-03751-f008:**
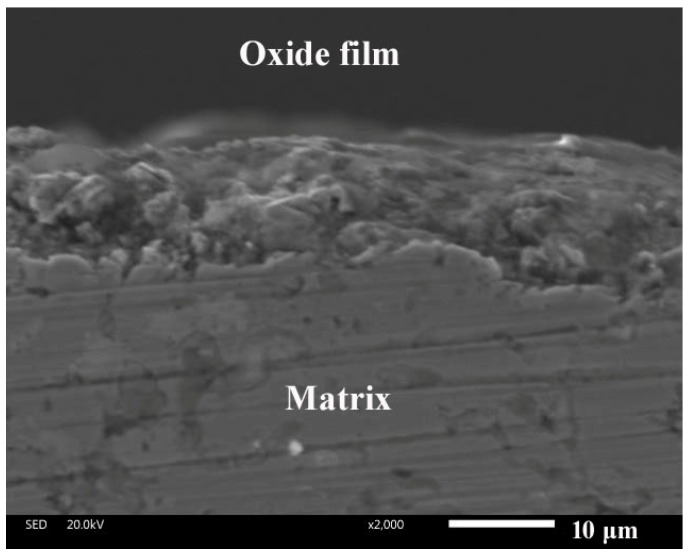
Oxide film on the wear track.

**Figure 9 materials-12-03751-f009:**
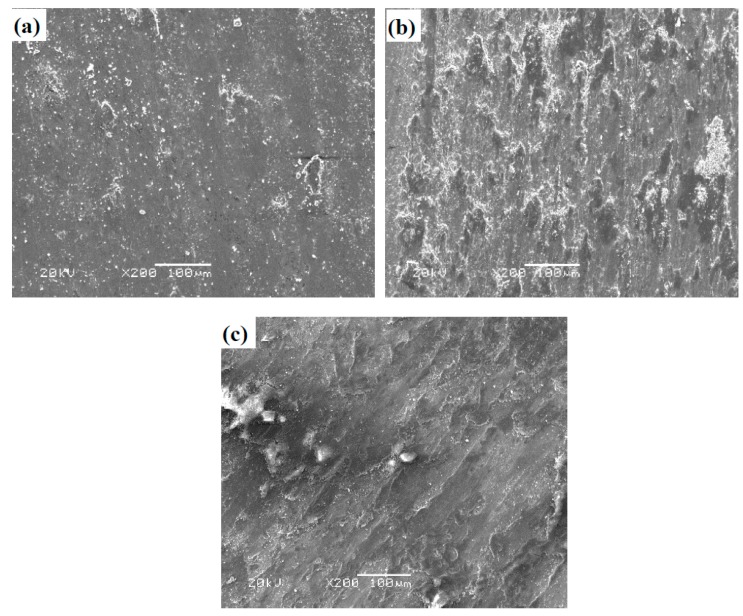
SEM images of worn surface morphologies of T: (**a**) room temperature (RT), (**b**) 600 °C, and (**c**) 900 °C.

**Figure 10 materials-12-03751-f010:**
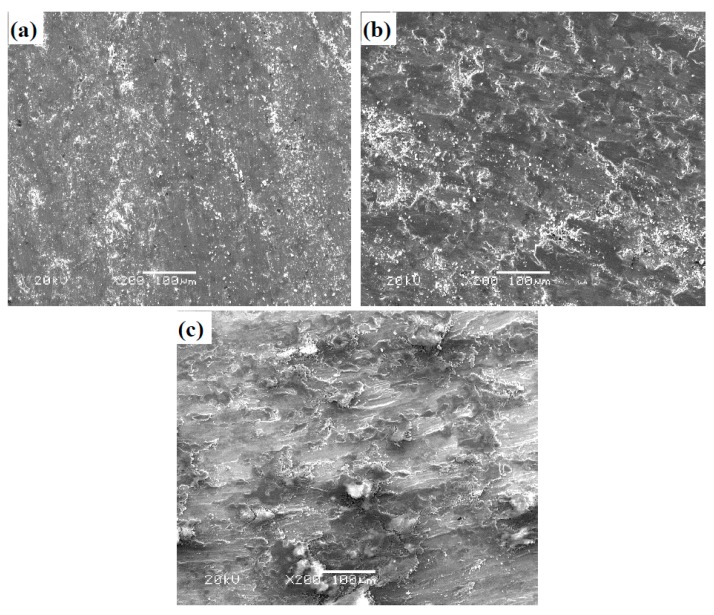
SEM images of worn surface morphologies of TB: (**a**) RT, (**b**) 600 °C, and (**c**) 900 °C.

**Figure 11 materials-12-03751-f011:**
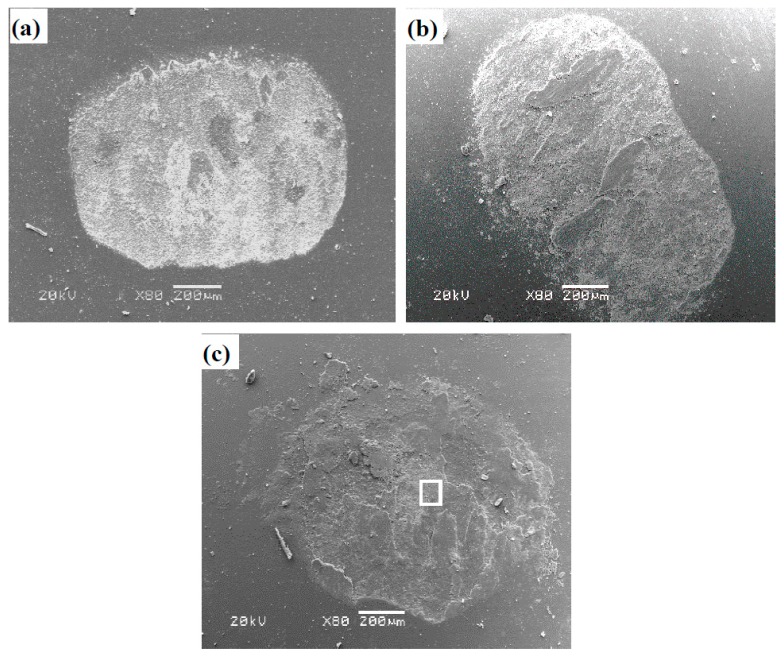
SEM images of worn surface morphologies of Si_3_N_4_ ceramic balls sliding against TB: (**a**) RT, (**b**) 600 °C, and (**c**) 900 °C.

**Figure 12 materials-12-03751-f012:**
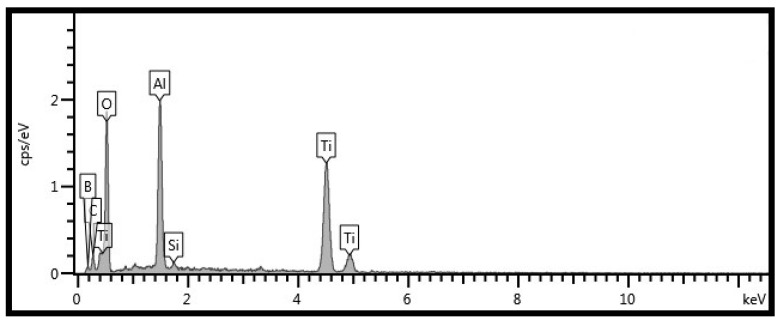
EDS analysis of marked area of Si_3_N_4_ ceramic ball in Figure 11c.

**Table 1 materials-12-03751-t001:** Composition of the pre-alloyed powders (wt.%).

Specimens	Ti	Al	B
T	78	22	0
TB	71	20.8	8.2

**Table 2 materials-12-03751-t002:** Mechanical and physical properties of specimens.

Specimens	*Hv* (GPa)	Compressive Strength (MPa)	Bending Strength (MPa)	Density (g/cm^3^)	Porosity (%)
T	6.71 ± 0.23	356 ± 9	317 ± 6	4.04	0.31
TB	8.97 ± 0.41	873 ± 8	535 ± 8	3.70	0.34

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
