# Peer review of "Preparation, Mechanical Properties, and High-Temperature Wear Resistance of Ti–Al–B alloy"

_materials, 2019, doi:10.3390/ma12223751_

Round 1
Reviewer 1 Report
Dear Authors,
The manuscript entitled “Preparation, Mechanical and High-Temperature Tribological Properties of Ti-Al-B Alloy" presenting Ti-Al matrix alloy reinforced with high boron element content fabricated by using high-temperature alloying method and powder metallurgy technique (P/M).
The topic is interesting, and I recommend it for publication after minor revision. Information are described clearly and are presented in the impressive form. The technique, technology and research methods used in work are adequate. Authors achieve the presented main goal of the work. Methods and obtained results prove founded thesis and show the originality of the manuscript. I think it will be an interesting paper for the readers.
Correctly cited figures and tables.
Literature is satisfactory.
However:
The information about the size of the powder was provided by the producer, or it was from measurements? If it was producer information, the company name should be presented. If it was a measurement, please provide a procedure and equipment how it was done. Ball on disc method is very common, so in my opinion, a graph of this method is not necessary. Table 2 the unit of HV is GPa instead of Gpa.
Moreover, the authors mentioned in table 2 about porosity, please provide information about how it was measured? The authors in section 2 wrote that fifteen tests of hardness were done, so please present an average, error and standard deviation of the hardness measurements. How many samples were made for compression test and three-point bending? If more than one, also the average error and STDV should be presented. Line 147 word “metallic” should be with lowercase letters.
Author Response
Dear Sir/Madam,
Thanks a lot for your perfect suggestions. These comments and suggestions are very helpful. We have revised our manuscript according to your comments and suggestions. The revised parts are marked in red in the present paper.
1. Question: The information about the size of the powder was provided by the producer, or it was from measurements? If it was producer information, the company name should be presented.
Answer: The sizes of raw materials had been added in the manuscript, and the company names had been given.
2. Question: Ball on disc method is very common, so in my opinion, a graph of this method is not necessary. Table 2 the unit of HV is GPa instead of Gpa.
Answer: Thanks for your suggestion. The graph was deleted.
3. Question: About porosity, please provide information about how it was measured?
Answer: The corresponding method is given in the manuscript. The porosity was calculated by the following formula: (see the manuscript)
where ρo was the theoretical density and ρ was the actual density of alloys.
4. Question: The authors in section 2 wrote that fifteen tests of hardness were done, so please present an average, error and standard deviation of the hardness measurements. How many samples were made for compression test and three-point bending? If more than one, also the average error and STDV should be presented. Line 147 word “metallic” should be with lowercase letters.
Answer: The hardness, compressive strength and bending strength of specimens are average value. Compression test and three-point bending of each specimens were repeated three times. The error had been added. Thanks for your suggestions.
Thanks for your patience.
Yours sincerely,
Gongjun Cui

Reviewer 2 Report
The work should strengthen the relevance of research. Focus on the promising direction of boron alloying with titanium, aluminum and other alloys, which are in demand in high-tech industries. The review should reflect the advantages of powder metallurgy over other methods, for example, compared to spark laser sintering or field assisted sintering technology (FAST). Consider wear resistant coatings. Be sure to include in the list of references articles:
Couret, A., Voisin, T., Thomas, M., Monchoux, J.-P. Development of a TiAl Alloy by Spark Plasma Sintering (2017) JOM, 69 (12), pp. 2576-2582. DOI: 10.1007/s11837-017-2549-6;
Sharma, V., Timmons, R.B., Erdemir, A., Aswath, P.B. Interaction of plasma functionalized TiO2 nanoparticles and ZDDP on friction and wear under boundary lubrication (2019) Applied Surface Science, 489, pp. 372-383 DOI: 10.1016/j.apsusc.2019.05.359;
Soliman, M.S., El Rayes, M.M., Abbas, A.T., Pimenov, D.Y., Erdakov, I.N., Junaedi, H. Effect of tensile strain rate on high-temperature deformation and fracture of rolled Al-15 vol% B 4 C composite (2019) Materials Science and Engineering A, 749, pp. 129-136 DOI: 10.1016/j.msea.2019.02.016;
As a result of this, change the title of the publication to more interesting for readers. For example, "High-temperature boron alloying in metal powder processes in the production of wear-resistant titanium alloys".
After correcting all the comments, the article will be more understandable and interesting for readers. It can be accepted into print.
Author Response
Dear Sir/Madam,
We quite appreciate your insightful comments and suggestions and have careful checked and revised our manuscript according to your comment and the replies are addressed in detail below. The revised parts are marked in red in the present paper.
Question: The work should strengthen the relevance of research and title.
Answer: Thanks for your suggestions. We have revised the relevance of research according to your suggestions. About the prepared methods of alloys, the powder metallurgy technique ensured the complete alloying of Ti, Al and boron element during a long sintering process. However, the heating rate of SPS is fast, which lead to the inadequate reaction.
Your suggested topics is very good. We have partly revised the title after discussion. Thanks for your suggestions.
Thanks for your patience.
Yours sincerely,
Gongjun Cui
Round 2
Reviewer 2 Report
The manuscript is better. And it can be accepted for print.